# Free School Meal Improves Educational Health and the Learning Environment in a Small Municipality in Norway

**DOI:** 10.3390/nu14142989

**Published:** 2022-07-21

**Authors:** Greta Heim, Ruth Olaug Thuestad, Marianne Molin, Asgeir Brevik

**Affiliations:** 1Department of Nursing and Health Promotion, Faculty of Health Sciences, Oslo Metropolitan University, NO-0130 Oslo, Norway; greta.heim@uit.no (G.H.); ruth.thuestad@gmail.com (R.O.T.); mmolin@oslomet.no (M.M.); 2Department of Health Sciences, Oslo New University College, NO-0456 Oslo, Norway

**Keywords:** school meals, educational health, learning environment, school environment, Norway, qualitative methods, in-depth interviews

## Abstract

It has been suggested that school meals could have an impact on students’ learning environments; however, existing research in this field is scarce and inconclusive. The purpose of this study was to explore teachers’ and school administrators’ experiences with the introduction of a free school meal and whether this influenced the learning environment. The study was conducted in upper primary and lower secondary schools in a small municipality in Norway. In this qualitative study, 17 informants participated in semi-structured in-depth interviews. The interviews were recorded, transcribed, and coded using NVivo. Thematic analysis was conducted using systematic text condensation (STC). The main findings are that in the informants’ experience, a free school meal led to reduced absenteeism during lunchtime and positive social interactions among students, social equalization, and a more peaceful atmosphere during lunchtime. In conclusion, the introduction of a free school meal had a positive impact on the students’ educational health and the learning environment, and contributed to social equalization as all the students shared the same healthy school meal.

## 1. Introduction

According to the World Health Organization (WHO) [1], schools should prioritize healthy food for students during the school day as it has a positive impact on students’ wellbeing and, thus, improves learning ability and academic performance. Both the WHO and the European Union emphasize the importance of promoting a healthy diet by focusing on food and meals in the school [2]. In line with these ambitions, the Norwegian Directorate of Health has developed a National Guideline for Food and Meals in Schools where the importance of the school meal for the learning environment is emphasized [3]. The aim of these guidelines is to ensure that students are offered a healthy meal and that there is a suitable structure for the meal.

While many countries in Europe have school meals that are entirely or partially funded by the government, the school meal in primary and secondary schools in Norway is, for the most part, based on the packed lunch brought from home. From an international perspective, the education and welfare systems are relatively similar in the Nordic countries, still, the school meal is organized in different ways. Finland and Sweden are in a unique position as they are required by law to serve free school meals for primary and secondary school students [4]. In Norway, as well as in Denmark, it is up to the school owner to decide if they want to offer a free school meal. Different practices in otherwise similar countries provide a good opportunity to compare different models. A systematic review of the literature on the effect of school meals on health and learning outcomes by the Nordic Council of Ministers concluded that knowledge about the long-term effect of school meals on learning is lacking [4].

The Norwegian Directorate of Education defines the learning environment as “the cultural, relational and physical conditions at school that influence students’ learning, health, and well-being” [5]. In a positive learning environment, students experience trust, recognition, feeling a sense of security, and being part of a fellowship and have a good relationship with the teacher [6].

School meals have already been subject to extensive study, but intervention studies carried out on school meals have mostly focused on food intake and food content [7], and the potential impact on social and psychological factors is much less studied. We wanted to explore whether offering a free school meal has an influence on the students’ learning environment. To examine this question, we asked teachers, school administrators and school owners in a small municipality in Norway their experiences of how the introduction of a free school meal affected students’ learning environments in primary and secondary school.

## 2. Materials and Methods

### 2.1. Study Design

As our goal was to gather the experiences and opinions of those involved with the introduction of a free school meal, an inductive approach was chosen. The method was individual, semi-structured in-depth interviews. The consolidated criteria for reporting qualitative research (COREQ) were used to enhance the transparency and quality of this study (Appendix A).

### 2.2. Setting

The municipality where this study was conducted is among a minority that offer free school meals in Norway. The municipal council decided to introduce a free school meal as a pilot project in 2013, and since then it has been a permanent offering at all of the three schools within this municipality. The municipality defines the school meal as part of their preventive health measures concerning diet and health. This was carried out based on a survey of living conditions, and the municipality allocated the necessary funding for offering a free school meal to students in primary and lower secondary schools. Our study includes upper primary schools, grades 5 to 7 and lower secondary schools, grades 8 to 10, which corresponds to students 10 to 16 years old. Further, the municipality council decided that the National Guideline for Food and Meals in Schools [3] would form the framework for the free school meal to be offered. Apart from these guidelines, the individual schools could decide how and when the free meal should be served. One of the schools chose to serve breakfast as the free school meal as many of their students live far from the school, thus they leave home early by school bus since this is a rural area. The other two schools have chosen to serve lunch. The free school meal is served as a buffet and consists of slices of bread with varied toppings such as ham, cheese, and light margarine, fruit and vegetables, as well as milk and water. The bread for the buffet is whole grain bread delivered from a home bakery. The smallest school also serves a hot meal to the students twice a week.

### 2.3. Participants

A strategic selection of 17 informants was included in the study. Nine of the participants were teachers from three upper primary and lower secondary schools, five were school administrators, two were school owners, and, finally, one was an ombudsman against bullying. For anonymization, the informants were coded with random letters and organized in two groups of teachers and administrators, where school owners and the ombudsman against bulling were included in the last group (Table 1).

The criteria for recruitment were that the informants had experiences working in the school system in the municipality prior, as well as, after the introduction of the free school meal. School owners were recruited by phone calls and subsequent letters. School administrators were recruited in co-operation with school owners, and teachers were recruited in cooperation with school administrators. In-depth individual interviews were conducted with the informants. Participation was voluntarily, and the informants gave their written informed consent before participation.

### 2.4. Data Collection

The in-depth interviews with the teachers and school administrators were conducted at the schools where they worked, and the school owners were interviewed at the town hall, in October 2019 by the two main authors (G.H. and R.O.T.) with 9 and 8 interviews, respectively. All interviews were conducted 6 years after the introduction of the free school meal at all the three schools.

The interview guide for the semi-structured interviews contained an overview of topics that could be covered and a suggestion for the order in which the questions should be asked. We started the interview with an introduction of the aim of the study and informed the participants that they could withdraw from the study at any time without any reason. Then we asked some questions about the informants’ background to create a comfortable atmosphere for the interview. The school meal was the next topic in the interview guide, with a focus on what may be different after the introduction of the free school meal. The last topic was related to the learning environment and their experiences with possible changes after the free school meal was introduced. In the process of making the interview guide, we visited the three schools and observed how the school meal was served. After making the first proposal for the interview guide, it was tested as a pilot at a school in another municipality. In this pilot, both the interview guide and the use of a dictaphone were tested before finalizing the interview guide [8]. Since the interview guide worked as intended, no changes were made after the pilot.

Dictaphones were used to record interviews. The transcription of the interviews was performed after all the interviews were completed.

### 2.5. Data Analysis

After the data collection, the names of the informants were anonymized using random letters from the alphabet. The interviewers (G.H. and R.O.T.) transcribed their own interviews. The NVivo QSR International 2020 software was used to guide the analysis of the collected data [9].

Systematic text condensation (STC) was chosen as the method of analysis. STC is based on a model from Malterud [10] and included the following four steps: (1) to get an overview of the collected data, both interviewers repeatedly read through all transcripts; (2) by filtering the data material, meaningful units were identified, and four preliminary themes were chosen; (3) organizing main themes into subthemes; and (4) recontextualization by rewriting the text to the third-person form to visualize someone else’s experiences.

The two interviewers discussed potential themes and subthemes, and after identification of thematic common denominators, original transcripts of the interviews were reread to ensure that records were in accordance with the informants’ opinions.

## 3. Results

Three distinct thematic domains emerged from the analysis of our results, each theme with two or three sub-themes (Table 2). The first main theme revolves around presence, with the sub-themes attendance, structure, and social relationships. The second main theme was equality with sub-themes of shared school meal and social equalization. The last overarching theme was peacefulness, with sub-themes of peaceful lunchtime and peacefulness for learning. In the presentation of the results, illustrative quotes are included.

### 3.1. Presence

The students must attend the school meal to experience a structured meal situation at school, and presence is required for social relationships to form.

#### 3.1.1. Attendance

Several of the informants said that the most significant change after the introduction of the free school meal was that the students from lower secondary school now actually chose to attend the school meal, whereas they previously left the school grounds during lunchtime to head to the town center to buy food. The following statement from one informant illustrates this:

“We have virtually no leave down to the center now, as opposed to what we had before we started with the free school meal.”Informant D.

Several informants stated that after free school food was introduced this changed so that now only one or two students leave the school regularly.

“We have stopped the traffic to the local store, as that was the lunch place (laughter). Yes, many went there before, now they rarely go.”Informant R.

These informants expressed that leaving the school grounds during the lunch break contributed to divide the student community, but when everybody was present at school during lunchtime it created a better school environment.

“I think that presence at school creates a better school environment. Being together when you have a break is important, and this can’t be accomplished if someone leaves.”Informant R.

Several informants experienced that if some of the students did not come back to school in time for the lesson after the lunchbreak, they decided to skip the rest of the school day.

“They went down to the store at lunchbreak and arrived late for the next lesson, or maybe not coming back at all, this happened particularly at the lower secondary school.”Informant M.

Furthermore, the informant said that this could apply to the disadvantaged students as it was difficult to get them back until the next day, but this problem dissipated quickly after introducing the free school meal. The same informant noticed that there was less absenteeism as the weaker students, as he expressed it, were more present in school. He also had a perception that shoplifting at stores had decreased. Many informants had the opinion that since the lower secondary students were more present at school, they had the opportunity to guide them in a more positive direction. The following quote illustrates this:

“Some students were absent and there were more factors that influenced them in everyday school life than desired. The fact that they often hung out in the store, and maybe someone could have negative experiences around the police because of…. yes…. shoplifting and so on. If you take this away and the students are present at the school, then at least the opportunity to influence them in a positive direction is considerably greater.”Informant V.

#### 3.1.2. Structure

Several of the informants had registered that the social structure in the canteen could be a challenge for some of the students after the introduction of the free school meal because the meal situation became more complex. One informant highlighted the challenge of uncertainty some students felt about finding a seat and whom to sit next to, and this insecurity seemed to emerge in those who lack a confident relationship with other students.

Some of the informants said that it was important that adults/teachers were present during the school meal both in primary and lower secondary school. This was due to them making sure that there was room for everyone by, for instance, adding chairs to the tables where the students wanted to sit. Some teachers said that they chose to sit together with the students to have lunch, even though they were not on supervision duty. This helped build relationships with the students, especially if there was a conflict or disturbance in the student group. One teacher said:

“The teachers get a different relationship with the students when they sit and eat together. Because then, in a way, you are not the teacher and do not stand and lead the class but sit and eat together and talk about topics from everyday life. So yes, I think it influences the relationship. Positive influence.”Informant W.

Two of the primary schools had chosen to have permanent seating during the school meal to avoid insecurity and dispute over seats. In this way, the students did not have to invite themselves to a table or be invited as they were a natural part of the group. At the largest school, they decided to have free seating for the school meal. Here, some informants expressed a wish for permanent seating to avoid the uncertainty and lack of confidence that may occur for some students. However, free seating still encourages new contacts and friendships across classes and age groups. Regarding this, some informants emphasized that this is important for the social life at school. One informant who supported free seating said:

“One can make friendships across age groups. I think this is important for the social environment and the feeling of fellowship with others.”Informant B.

#### 3.1.3. Social Relationships

Many believed that if lunch time has been used for relationship building it would be transferred into the classroom and, subsequently, create a better learning environment. One informant mentioned that the school meal is a great way to create social relationships between students because it also includes those who have problems, or who, for example, are shy and fear speaking up. The social aspect is important for everyone, but for these students, it is even more important. Some believed that the free school meal promotes mental and physical health and well-being.

“The fact that they are social during the school meal influences their well-being. It shows that they can talk to whom they want around the table, and you can see that they thrive together. I also believe that the way we organize the school meal helps prevent bullying because all the students are together during the school meal and get to know each other.”Informant P.

One informant also mentioned that since this is a rural municipality, the geographical distances within the municipality are quite large, and some of the students do not have the opportunity to visit friends after school. For these students, the social aspect of the school meal is very positive and important for building relationships and making friends. Thus, the social meal situation allows the students to create a network.

One of the schools had intentionally chosen round tables in the canteen enabling everyone to sit in a circle looking at each other. They thought it was important for the students to be seen and to feel a sense of belonging. Several believed that the students experienced the school meal as something safe, secure, and predictable. The students have time to be together in companionship, and it is important for their psychosocial environment. All the informants expressed that by mixing the age groups at lunchtime the students got the opportunity to create friendships across age levels, which is important for the social environment and a sense of companionship. Some teachers reported that they have students who do not have meals together with their families at home, and therefore it is even more important to have a common school meal and gather around the tables to eat and talk. Many expressed that they believed that this would provide social development, which may be lacking in some homes because the family is too busy in everyday life to sit down and have a meal together.

### 3.2. Equality

The fact that everyone eats the same food in the shared school meal can help reduce the social inequalities among students.

#### 3.2.1. Shared School Meal

The municipality serves a free school meal to all the students. The informants emphasized that since all the students share a common meal, they have eliminated the food as the cause of differences among students.

“It is a different calmness, because I think that for this age group it is important to be equal to their peers, and this is what they want to be. When they are offered the same food, they can relax, they do not have to think, oh, what do you bring and what do I bring in the lunch box, and then it is easier to calm down. They can enjoy themselves. They smile and laugh, and they chat with each other, and it seems like they are having a great time around the tables.”Informant L.

Several informants emphasized that after the free school meal was introduced, there was no difference between the students who had a packed lunch and the students who did not bring a packed lunch from home. The majority of the informants from the lower secondary level said that there were very few students who brought a packed lunch from home before the free school meal was introduced. Some informants pointed out that through the school meal, the students are guaranteed at least one consistent meal during the day. Many expressed that after the introduction of free school meals, there were fewer students who were hungry and tired at the end of the day.

#### 3.2.2. Social Equalization

Many believed that the greatest and most important difference after introducing the free school meal was the social equalization that was unique in relation to the students’ different backgrounds. One informant described the situation where the children sat around the table, having a great time together, regardless of their family background. Several expressed that free school meals had a unifying effect on inclusion, as bringing packed lunch from home could be exclusionary.

“The school meal with packed lunch is partly exclusionary. When it is a quarter to eleven, and the teacher asks everyone to bring out the packed lunch, and then you realize that you do not have a packed lunch! With the free school meal, this is equalized. Then you have the social factor of eating in groups, which is organized differently than before. In the past, the students sat in the classroom with only the classmates and ate, while now they like to mix classes.”Informant M.

The experience of building companionship across social backgrounds increases school well-being and provides a better school environment.

“The biggest difference is the social equalization you get with everyone getting the same food. That’s number one. Number two is the learning effect of sitting around a table and having a conversation with a classmate, or a friend.”Informant V.

The same informant had experienced that not all students had the same support at home, as some parents do not think school is important or are not as concerned with teaching the children healthy eating habits. Another informant thought that it was important to think holistically, and that having a healthy meal in a social setting is something that not everyone experiences at home.

“It’s a value they will carry on for the rest of their lives, eating healthy, nutritious food together with others. I think it has even more significance in the future than just for the time being.”Informant W.

Some believed that it has significance also for the future since students incorporate good eating habits and can concentrate while they are in the school. This may influence their education and job opportunities in the future. Some highlighted that this is preventive work, also in terms of competence, because in primary school the foundation for a good life is created. An informant mentioned that the financing of the school meal is money well spent for the municipality, as she expressed it:

“The money the municipality spends on the school meal is money they might save in the long run, because in the future these students, as adults, can function better, be healthier, work more. We will not know the total effect of the free school meal until some years have passed.”Informant I.

The established eating habits can affect lifestyle diseases, such as cardiovascular disease and diabetes. One informant pointed out that the municipality has had problems with dropouts from high school and many receive social financial support.

“I think it’s good for physical health too because we have healthy and predictable meals. They get food, it’s healthy, and I think that’s good for them. Also, in the future, of course, they get healthy eating habits, they can concentrate on schoolwork, and learn more. In a way, they lay the foundation for a good life.”Informant I.

Many informants at the upper primary level stated that before introducing the free school meals, their students brought a packed lunch from home. The large group was, in a way, similar; however, some stood out, and some of the informants thought that it was related to socioeconomic conditions. They noticed that the socio-economical differences have been reduced. Many expressed that it was important that everyone got the same opportunities, at least at school.

“You get social equalization that is unique considering backgrounds. You have those who come from low social status together with those from high social status around the same table and have the same opportunities for eating food and having a good time together. We see that this has an effect also with companionship, school well-being, and in relation to the learning environment.”Informant V.

### 3.3. Peacefulness

The category of peacefulness emerged from the data material at an early stage. It was related to a more peaceful atmosphere during lunchtime as well as in the classroom after the school meal.

#### 3.3.1. Peaceful Lunchtime

Many thought the atmosphere was more peaceful in the canteen after the introduction of the free school meal, and that the actual eating situation was calmer. This is described in the quote:

“When you enter the canteen, it is a peacefulness, it is such a pleasant atmosphere.”Informant V.

The informants experienced that the students ate more during lunchtime because all the students had to take a united lunchbreak for a minimum of 15 min. This meant that they could lower their shoulders and relax as they all had to be together for the common meal.

“And then students sit very calmly and quietly and talk together around a table and eat their food in a very nice way.”Informant V.

#### 3.3.2. Peacefulness for Learning

One informant described the change after the introduction of the free school meal by this quote:

“It is more peaceful in the classroom. It is hard to describe it, because it is like a…, yes there is a completely different calmness around the lessons than it was previously. Completely different calmness.”Informant E.

Many experienced that there was a more peaceful atmosphere in the classroom and that when the students entered the classroom there was a readiness to learn. The students had already entered learning mode, which is described in the following quote:

“The free school meal is very important for the learning environment because it evens out these differences we have had here before. It creates peace for all the students, and those who were perhaps a little restless and who used to go outside the school area in the lunch break, they’ve been in school in recess and enter the classroom together with the others, so it creates a different calmness also when we start the lesson.”Informant E.

Many teachers thought that the students now had a better ability to concentrate and learn, and that they worked harder. Another factor that several pointed out was that some students were less aggressive, and that they did not give up as easily as before when they faced challenging issues. Some informants believed that if the students ate unhealthy food it led to more unrest and distraction, particularly at the end of the school day. This created challenges, especially among the students who already had a restlessness in the body, as they expressed it. Because of the conversations with the students during lunchtime, the teachers felt that they came closer to the students, and since they had a better relationship, they expressed that it was easier to communicate with the students in the classroom as well.

## 4. Discussion

By being offered a free school meal, the informants experienced increased attendance at the school meal, and consequently the students built better social relationships. The fact that everyone ate the same food was experienced to have a socially equalizing effect. Since all the students participated in the free school meal, the informants experienced that this generated a more peaceful atmosphere during lunchtime as well as in the classroom.

### 4.1. Presence

One of the most important findings in this study was that the students did not leave the school grounds during lunchtime after the introduction of the free school meal as occurred previously. Informants believed that the act of sharing a common meal, with everybody present, provided the students with the opportunity to create relationships with other students, and with the teachers attending. This confirms the assumption that the free school meal contributes to a better learning environment [11]. However, our results are in contrast with the results found in the school food project in Aust-Agder in Norway [12], which concluded that the free school meal did not affect how the students perceived the classroom environment. While the free school meal in our study had been established for 6 years, this study was conducted after 5 months, which may not be enough time to recognize long-term changes in the learning environment.

The municipality experienced that shoplifting at the nearby shops decreased after the introduction of free school meals, and this was further supported by our informants who experienced that minor crime immediately decreased. Some specifically mentioned that if students did not show up for class or arrived late for class, it may have been the beginning of dropping out of school. They believed that a consequence of a free school meal might be that more students would be able to complete upper secondary school and, subsequently, higher education. This is supported by Cueto [13] who concluded that the school meal has a positive effect on attendance and increases the probability of students completing their education. In a systematic review of 47 studies, it was found that nearly all studies examining free school meal reported positive associations between school meal participation and academic performance [14].

According to “Ungdata”, a national survey in Norway which is regarded as the most comprehensive source of information on adolescent health and well-being, the percentage of absenteeism for the municipality in our study was 18% [15]. In comparison, three of the nearby municipalities of the same size had 34%, 27%, and 23% absenteeism, and the nearest town had 36% absenteeism, while the national average was 24%. Interestingly, the same survey from 2013 [16], which was conducted before the free school meal was introduced in the municipality where this study was conducted, showed 29% absenteeism in this municipality, which at that time was higher than the national average of 23%. Hence, there has been a 40 percent reduction in absenteeism following the introduction of the free school meal, although this may be a coincidence, one cannot rule out that this may be due to the introduction of the free school meal.

Based on informants’ statements, we understand that the school meal is a social arena, and the results show that there is substantial social learning during the school meal. The development of social competence is an important part of education [17]. The Norwegian Directorate of Health [18] insists that there should be a good structure to the school meal. Several informants emphasized that the school meal is the foundation for a positive learning situation.

The school meal was organized outside the classroom in a canteen setting for all three schools in this study. Our informants saw this as a great advantage as this provided an optimal arena for relationship building. We interpret that both the students and the teachers experienced the canteen to be a somewhat informal and relaxed arena which could be associated with going to a café with some friends. This setting could contribute to creating positive relationships between students, and between teachers and students. This is in accordance with findings from the School Food Project in Norway in the region of Trøndelag, which showed that the school meal formed the structure for socialization [19]. In addition, it corresponds with the Finnish authorities who believe that the school meal is part of the success of the school system in Finland [20].

WHO emphasizes that schools must facilitate access to healthy food during the school day, and maintains that the school meal has a positive impact on the well-being of students, which in turn leads to a better learning environment [1]. Decent dietary habits, attitudes, and behaviors can be formed by sharing a school meal [21].

After evaluating the project with free school meals, the municipality discovered other improvements in the education sector with the students developing a better relationship with each other. Again, when comparing data from the Ungdata survey [15], we see that in the municipality with the free school meal, 4% of students stated that they were being bullied, while in the three nearby municipalities about the same size, 8% state that they were being bullied. In the nearest urban municipality, the level is 9% [15]. This may be due to coincidence; however, sitting at round tables and having a common lunch with an increased presence of adults creates good relationships where everybody is included. This seems to result in less bullying and, consequently, an improved learning environment.

### 4.2. Social Equalization

Overall, findings show that being equal is important to students, and by serving everybody the same healthy lunch students became more similar, and the informants experienced social equalization. This observation applied to all grade levels, even though teachers in upper primary school said that their students generally brought healthy packed lunches before free school meals were introduced, while in lower secondary school there were many students without or with unhealthy lunch packages. The observations of our informants are in accordance with Nordahl [22], who concluded that the introduction of free school meals will have the greatest significance for the lower secondary school students and will not have a major impact on primary schools.

Vik’s [23] study showed that by serving a free school meal to 10–12-year-olds, children’s intake of healthy food increased, especially among children from families with lower socioeconomic status. A survey by Samdal [24] showed that there was a change when students entered lower secondary school, as many students, especially boys from families with low socioeconomic status, did not bring a packed lunch and, hence, did not always take part in the school meal. In a qualitative study on students’ participation in a free school meal in Oslo, Norway, Mauer et al. [25] found that offering a free school meal did not necessarily imply that the students took advantage of the offer. They found that the popularity of the food served, competition with the nearby shopping center, and social aspects where factors that influenced participation. In a cross-sectional survey in Norway where the students were served a hot school meal every day for 10 days, they found that even though the students wanted free school meals, several did not like the dishes served [26]. However, both students and school personnel experienced the meals as positive for the social environment.

The experience of the informants in our study was that after the free school meal was offered to all the students in the municipality, almost 100% of the students participated. This contrasts with our neighboring countries, Sweden and Finland, where many students were absent and chose to not take advantage of the free school meal. Although a free three-course hot school meal are served in Finland, not all the students eat this meal, or they only eat part of it. In Finland, only around 25% of girls and 33% of boys consume the whole school meal. Furthermore, 20% of girls and 12% of boys stated that they rarely or never eat the main course [4]. In Sweden, research shows those with low socioeconomic status took advantage of the offer more frequently than the average [4]. In two different school food programs in the USA, less than 60% participated [27]. Our results contrast the results in these countries, as the informants in our study stated that between 95% and 100% of the students took advantage of the free school meal every day. This may be due to less competition with alternative food sources for the students in our study, as it takes time to reach the nearest food store from the schools.

Several informants emphasized that being offered nutritious food in a common lunch helped to even out the social differences between the students. Florence [28] concluded that there was a connection between eating nutritious food and school performance. To establish healthy eating habits early in life is important because it influences both the health and the school performance of individuals, and in the future, it could have an impact on which socioeconomic status the students will have. In The Norwegian Public Health Report [29] two of the main goals are to reduce social health inequalities and to promote health in the entire population. To achieve this, the school is highlighted as an important health-promoting arena. In our study, several of the informants emphasized the importance of social equalization achieved by everybody eating the same free school meal, and several believed that the social differences had become less noticeable. This is supported by the results of a longitudinal study in Sweden, where it was found that the free school lunch generated long term benefits, where primary school students participating in the program had 3% higher lifetime income [30]. The effect was greater for students from poor households, suggesting that free school meals reduce socioeconomic inequalities in adulthood.

### 4.3. Peacefulness

All the informants agreed that the introduction of the free school meals led to a special peaceful atmosphere in the eating area. The informants also emphasized that it is important for adults to be present during the meal, and they saw the presence of adults as crucial for the calm atmosphere to be generated. The school meal-induced peacefulness extended to the classroom, according to the informants. Similar results have been reported elsewhere. In a qualitative study in a school food project in Trøndelag in Norway [19], the teachers believed that the school meal had a positive impact on the learning environment in the classroom. For the troublemaking students, who previously negatively impacted the learning environment for the whole group, the change was noticeable. Thus, offering a free school meal resulted in an improved learning environment for all.

The informants experienced that since all the students participated in the free school meal, this became a natural platform to make new relationships that eventually could result in new friendships. Since the school meal was organized in such a way that the students had to relate to others beyond their inner circle of friends, we believe that the students became more inclusive and generous. Other studies have shown that students who thrive and have self-esteem experience a good learning environment [31]. It stands to reason that a positive learning environment could reduce the importance of the parents’ level of education and minority status as a determinant for the student’s development [32]. According to a controlled intervention study by Golley et al. [33] which was conducted on lower primary school students, it was found that the school meal affected student behavior in the learning situation, to some extent. This is supported by Storey [34], who in a randomized controlled trail (RCT) of lower secondary school students, concluded that by facilitating the food offering and the eating environment for the school meal, it was possible to improve the learning environment. Our findings correspond with Bellisle [35], who concluded that eating nutritious food on a regular basis causes young people to function well, both in terms of learning and behavior.

### 4.4. Strengths and Limitations of the Study

Direct information from those who experienced the school meal through their professional work represents a strength of this study. With 17 interviews, this study has a reasonable number of respondents. We believe the inherent structure and guidelines in STC have strengthened the study’s reliability and validity. The fact that the informants had good communication skills was an advantage during the in-depth interviews. By choosing individual in-depth interviews rather than focus group interviews, we let each informant come forward with their own experiences and thoughts without being influenced by others.

A limitation of the study may be that the transcription was not performed immediately after each interview. For practical reasons, all the interviews were completed before we started the transcription. Each of the primary authors (G.H. and R.O.T.) transcribed their own interviews, and according to Kvale and Brinkmann [36] this is a strength as the transcription could then, to a certain extent, be characterized by the actual situation during the interview, both in terms of social and emotional aspects.

The results of this study are expected to be transferable to, for example, other schools in rural municipalities in Norway. Further research on how the learning environment is affected by a free school meal is needed, as our study was conducted on a limited selection of upper primary and lower secondary schools in a rural municipality in Norway.

### 4.5. Educational Health

Educational health is determined by a combination of the physical-, psychological-, and social learning environments: Based on our results, we would like to suggest the term “educational health” as a top-level denomination that incorporates the physical, psychological, and social learning environment domains. To summarize our findings, we have made a figure to illustrate how educational health is influenced by the different segments of the learning environment (Figure 1). The figure is inspired by Benn [37], who divides the learning environment into three different segments: The physical learning environment, the psychological learning environment, and the social learning environment.

The physical learning environment includes being present during the school meal. This can affect the psychological learning environment, which is about the students, themselves, being able to choose what they want to eat and drink within the selection that is the same for everyone, as well as the meal increasing the chance of concentration regardless of socioeconomic background. This, in turn, can affect the social learning environment, which includes the structure during the meal, and provides peacefulness and opportunities to form positive relationships. The different segments of the learning environments are connected and will have an impact on the overall learning environment at the school.

If we reflect on the WHO’s [38] definition of health, as “a state of complete physical, mental, and social well-being and not merely the absence of disease or infirmity”, the different learning environments could affect students’ health. The physical learning environment enables students to be physically healthy by eating nutritious food. The psychological learning environment can affect the mental health through the student’s sense of participation and self-esteem. The social learning environment can affect social health when students thrive and have the opportunity to form positive relationships with other students at the school meal.

For optimal educational health there needs to be a balance of the physical-, psychological-, and social learning environments. A lack in any of these three domains creates suboptimal educational health. Compared to the socioeconomic model, optimal educational health can be of individual significance for the students in the short- and long-term, and can have an impact on the social status the student will acquire in the future.

## 5. Conclusions

In this study we found that the informants experienced positive changes in the learning environment after a free school meal was introduced. Many of the informants described that the introduction of a free school meal had the effect that the students who previously left the school area during lunchtime now participated in the shared school meal. With increased attendance during lunchtime, the ground is prepared for stronger relationships to be formed between students, as well as between students and teachers. Several of the informants described social equalization as a result of the free school meal program. Many of the informants experienced a more peaceful lunchtime, and this, they claimed, generated an atmosphere more conducive to learning. Several of the informants reported a noticeable improvement in the learning environment following the introduction of the free school meal, particularly in lower secondary school.

Our results demonstrate that the introduction of a free school meal can have a positive impact on educational health and the learning environment. As our study is based on a limited selection of upper primary and lower secondary schools in a rural municipality in Norway, we suggest a nation-wide follow-up study on the association between free school meals and the learning environment in primary and secondary schools.

## Figures and Tables

**Figure 1 nutrients-14-02989-f001:**
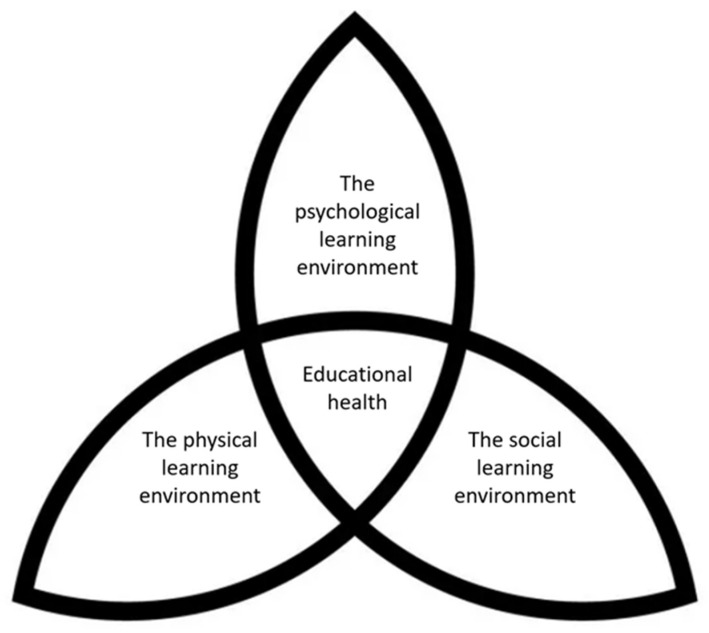
Educational health is influenced by the physical-, psychological-, and social learning environments.

**Table 1 nutrients-14-02989-t001:** Codes for informants.

Informant	Code
Teachers	B, D, E, I, O, P, Q, R, Y
Administrators	F, G, H, L, M, U, V, W

**Table 2 nutrients-14-02989-t002:** Main themes and sub-themes.

Main Themes	Sub-Themes
Presence	○ Attendance ○ Structure ○ Social relationships
Equality	○ Shared school meal ○ Social equalization
Peacefulness	○ Peaceful lunchtime ○ Peacefulness for learning

## Data Availability

The anonymized transcribed interviews are available from the corresponding author.

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
