# Peer review of "Free School Meal Improves Educational Health and the Learning Environment in a Small Municipality in Norway"

_nutrients, 2022, doi:10.3390/nu14142989_

Round 1
Reviewer 1 Report
This paper gives a chance considering the role of school meal on improvement of educational health and learning environment.
This may be valuable as a business report rather than as a scientific paper.
The expressions some and /or many used in the results section should be replaced to the specific numbers (percentages).
Author Response
Comments and Suggestions for Authors
This paper gives a chance considering the role of school meal on improvement of educational health and learning environment.
This may be valuable as a business report rather than as a scientific paper.
The expressions some and /or many used in the results section should be replaced to the specific numbers (percentages).
Response:
Thank you for the response. However, since qualitative method is used in this study, we don’t think it is appropriate to use percentages and specific numbers to express the results.
Reviewer 2 Report
Specific review comments are below:
1. Page 1, line 31—In line with these ambitions The Norwegian Directorate of Health has developed…..
Suggest revising the “T” for the “The” to lower case “t” as “the”
2. Page 2, lines 74-75 – Our study includes upper primary schools; grades 5 to 7, and lower secondary schools; grades 8 to 10, which corresponds to students 10 to 16 years old.
Please clarify what authors mean by the “upper primary school” and “lower secondary school” here. Is it classified based on the median household income or based on geographic location of the school?
3. Page 2, lines 79-81 – One of the schools chose to serve breakfast as the free school meal……. The other two schools have chosen to serve lunch.
In this study, a total of three schools are selected. How do those selected schools represent the schools’ profiles in the same region? Are they similar in students’ demographics, grades, number of students, and number of teachers? Please provide information on the sampling representation in the “Setting” section or in the “Discussion” section.
4. Page 3, line 98 – In-depth interviews were conducted…
Suggest modifying the sentence to “In-depth individual interviews were conducted…”
5. Page 3, lines 111-112 – The school meal was the next topic in the interview guide, with a focus on what may be different after the introduction of the free school meal.
Schools with a longer history of offering free meals may have different experiences compared to schools just started to offering free meals. Please provide additional information related to when the interviews were conducted after the free school meals were offered for each of the selected sample schools.
6. Page 3, lines 117-118 – In this pilot, both interview guide and the technical equipment were tested before finalizing the interview guide.
Please give an example for the technical equipment used during the pilot test. Was the same technical equipment used in this study?
7. Page 3, line 125 – The NVivo software was used to guide the analysis of the collected data.
Please provide the specific version of the NVivo software was used.
8. Page 8, lines 368-370 – The informants believed that when the students were present at lunchtime and had a common meal, both the students and the students and teachers had the opportunity to create relationship.
Suggest revising the sentences to “… both between the students or the students and teachers has the opportunity to create….”
9. Page 8, line 372 – However, our results are in contrast with the results found in the school food project in … Page 8, line 375 – … meal in our study had been established for 6 years, the intervention period for this study ….
Do authors mean that the “school food project” is the intervention study? Please clarify.
10. Page 9, lines 394-395 – …., which was done before the free school meal was introduced in the municipality were this study was conducted,….
Do author mean the word “were” for “where”?
11. Page 9, line 397 -- .. has been a 40 per cent.
Please revising the “per cent” to one word “percent.”
12. Page 9, line 439 – Vik (23)’s study showed that by serving a free school meal to 10—12-years olds, …
Suggest revising this sentence to “Vik’s (23) study showed that by serving a free school meal to 10-12 years old, ….”
13. Page 10, line 457 -- , not all the students eat thismeal, or they only…..
Please break the word “thismeal” into two words “this meal”
14. Page 11, lines 504-505 – This is supported by Storey (343) who in an RCT of lower secondary school students concluded that by facilitating…..
Please clarify what does the “RCT” mean here.
15. Page 12, line 560 – Compared to the Sosio Economic Model, optimal education…..
Please clarify what does the word “Sosio” mean.
16. Page 13, Table 1 – Columns “Item” and “Guide questions/description”
Please switch these 2 columns for easier context reading.
17. Page 15, item no. 14. – Setting of data collection: All interviews were collected at the participants workplace.
This description seems different than what is stated under “Data Collection” on Page 3, line 102-104 – The in-depth interviews with the teachers and school administrators were conducted at the schools where they worked, and the school owners were interviewed at the town hall…..Please clarify.
18. Page 15, item no. 22. Data saturation: Data saturation was achieved.
What is the definition of data saturation in this study? Please add data saturation information in the data analysis section.
Author Response
Comments and Suggestions for Authors
Specific review comments are below:
- Page 1, line 31—In line with these ambitions The Norwegian Directorate of Health has developed…..
Suggest revising the “T” for the “The” to lower case “t” as “the”
Response:
Text is changed as suggested: Page 1 line 33:
In line with theses ambitions the Norwegian Directorate of Health has developed….
- Page 2, lines 74-75 – Our study includes upper primary schools; grades 5 to 7, and lower secondary schools; grades 8 to 10, which corresponds to students 10 to 16 years old.
Please clarify what authors mean by the “upper primary school” and “lower secondary school” here. Is it classified based on the median household income or based on geographic location of the school?
Response:
In the international school system, primary school is divided in lower primary school and upper primary school. Secondary school is divided in lower secondary school and upper secondary school. Since our study was done in grades 5 to 7, it corresponds with the international term of lower primary schools, and grades 8 to 10, which corresponds with the international term of lower secondary schools. In light of this, we don’t think it is necessary to clarify this in the text.
- Page 2, lines 79-81 – One of the schools chose to serve breakfast as the free school meal……. The other two schools have chosen to serve lunch.
In this study, a total of three schools are selected. How do those selected schools represent the schools’ profiles in the same region? Are they similar in students’ demographics, grades, number of students, and number of teachers? Please provide information on the sampling representation in the “Setting” section or in the “Discussion” section.
Response:
In this Municipality there are only 3 schools in total, and all 3 schools are included in this study.
To clarify this, we will add this the text on Page 2 line 69:
The municipality council decided to introduce a free school meal as a pilot project in 2013, and since then it has been a permanent offer at all the three schools within this municipality.
- Page 3, line 98 – In-depth interviews were conducted…
Suggest modifying the sentence to “In-depth individual interviews were conducted…”
Response:
Text is changed as suggested: Page 3 line 95
In-depth individual interviews were conducted….
- Page 3, lines 111-112 – The school meal was the next topic in the interview guide, with a focus on what may be different after the introduction of the free school meal.
Schools with a longer history of offering free meals may have different experiences compared to schools just started to offering free meals. Please provide additional information related to when the interviews were conducted after the free school meals were offered for each of the selected sample schools.
Response:
Added text: Page 3 line 101:
In-depth interviews with the teachers and school administrators were conducted at the schools where they worked, and the school owners were interviewed at the town hall, in October 2019 by the two main authors (G.H. and R.O.T.) with 8 and 9 interviews respectively. All interviews were done 6 years after introduction of the free school meal at all the three schools.
- Page 3, lines 117-118 – In this pilot, both interview guide and the technical equipment were tested before finalizing the interview guide.
Please give an example for the technical equipment used during the pilot test. Was the same technical equipment used in this study?
Response:
The equipment was Dictaphone, and the same Dictaphone was used in the study.
The word “technical equipment” is changed to “Dictaphones” on Page 3 line 111 and 112:
In this pilot, both interview guide and the use of Dictaphone were tested before finalizing the interview guide.
- Page 3, line 125 – The NVivo software was used to guide the analysis of the collected data.
Please provide the specific version of the NVivo software was used.
Response:
Add the specific NVivo software: Page 3, line 119:
The NVivo software QSR International 2020 was used to guide the analysis of the collected data.
- Page 8, lines 368-370 – The informants believed that when the students were present at lunchtime and had a common meal, both the students and the students and teachers had the opportunity to create relationship.
Suggest revising the sentences to “… both between the students or the students and teachers has the opportunity to create….”
Response:
Text is changed: Page 8 and 9 line 326-329:
Informants believed that the act of sharing a common meal, with everybody being present, provided students with an opportunity to create relationships with other students and with the teachers attending.
- Page 8, line 372 – However, our results are in contrast with the results found in the school food project in … Page 8, line 375 – … meal in our study had been established for 6 years, the intervention period for this study ….
Do authors mean that the “school food project” is the intervention study? Please clarify.
Response:
Page 9 line 331: “the intervention period for” is removed from the text, and changed to:
……..While the free school meal in our study had been established for 6 years, this study was done after 5 months, which may not be enough time to recognize long term changes in the learning environment.
- Page 9, lines 394-395 – …., which was done before the free school meal was introduced in the municipality were this study was conducted,….
Do author mean the word “were” for “where”?
Response:
Text is changed as suggested: Page 9 line 345:
….which was done before the free school meal was introduced in the municipality where this study was conducted….
- Page 9, line 397 -- .. has been a 40 per cent.
Please revising the “per cent” to one word “percent.”
Response:
Text is changed as suggested: Page 9 line 347:
.. has been a 40 percent….
- Page 9, line 439 – Vik (23)’s study showed that by serving a free school meal to 10—12-years olds, …
Suggest revising this sentence to “Vik’s (23) study showed that by serving a free school meal to 10-12 years old, ….”
Response:
Text is changed as suggested: Page 10 line 380:
“Vik’s (23) study showed that by serving a free school meal to 10-12 years old, ….”
- Page 10, line 457 -- , not all the students eat thismeal, or they only…..
Please break the word “thismeal” into two words “this meal”
Response:
Text is changed as suggested: Page10 line 393:
…., not all the students eat this meal, or they only…..
- Page 11, lines 504-505 – This is supported by Storey (343) who in an RCT of lower secondary school students concluded that by facilitating…..
Please clarify what does the “RCT” mean here.
Response:
Text changed: Page 11 line 432:
This is supported by Storey (343) who in a randomized controlled trail (RCT) of lower secondary school students concluded that by facilitating…
- Page 12, line 560 – Compared to the Sosio Economic Model, optimal education…..
Please clarify what does the word “Sosio” mean.
Response:
Text changed: Page 12 line 477:
Compared to the socioeconomic model, optimal education…..
- Page 13, Table 1 – Columns “Item” and “Guide questions/description”
Please switch these 2 columns for easier context reading.
Response:
Thank you for the response. However, since the original COREQ checklist is constructed this way, we just followed the guideline given there.
- Page 15, item no. 14. – Setting of data collection: All interviews were collected at the participants workplace.
This description seems different than what is stated under “Data Collection” on Page 3, line 102-104 – The in-depth interviews with the teachers and school administrators were conducted at the schools where they worked, and the school owners were interviewed at the town hall…..Please clarify.
Response:
This is correct as teachers and school administrators worked at the schools, and the in-depth interviews with them were conducted at the schools where they worked, and the school owners were interviewed at the town hall where they worked.
- Page 15, item no. 22. Data saturation: Data saturation was achieved.
What is the definition of data saturation in this study? Please add data saturation information in the data analysis section.
Response:
Page 15, Item no 22: Text changed to:
Data saturation was not discussed.
Reviewer 3 Report
This paper uses in-depth interview data to identify the positive impact of free school meals on children. As the authors point out, little research has been done on the various effects and evaluations of school meals. Hence, this paper is of academic value.
However, in the country where the reviewer lives, school meals have been offered in almost all elementary school and many secondary schools for quite some time (although there is a fee for this service except for low-income families); therefore, the points raised in this paper did not seem particularly new to me. In addition to the points made in this paper, other benefits in terms of nutritional balance have also been pointed out in my country. However, this paper does not discuss the benefits in terms of nutritional balance. Given the nature of the journal (Nutrients), it seems to me that the impact of free school meals on nutritional balance for school children should also be discussed.
Author Response
Comments and Suggestions for Authors
This paper uses in-depth interview data to identify the positive impact of free school meals on children. As the authors point out, little research has been done on the various effects and evaluations of school meals. Hence, this paper is of academic value.
However, in the country where the reviewer lives, school meals have been offered in almost all elementary school and many secondary schools for quite some time (although there is a fee for this service except for low-income families); therefore, the points raised in this paper did not seem particularly new to me. In addition to the points made in this paper, other benefits in terms of nutritional balance have also been pointed out in my country. However, this paper does not discuss the benefits in terms of nutritional balance. Given the nature of the journal (Nutrients), it seems to me that the impact of free school meals on nutritional balance for school children should also be discussed.
Response:
Thank you for the response. In Norway very few elementary and secondary schools offer free school meals, and the benefits of such an offer is debated. The purpose of this study was not to discuss the impact of free school meals on the nutritional balance for children, but rather to explore whether offering a free school meal has an influence on the students learning environment. Thus, describing the nutritional balance of the school meal was not the aim of our study.